# Multi-Task Learning-Based Deep Neural Network for Steady-State Visual Evoked Potential-Based Brain–Computer Interfaces

**DOI:** 10.3390/s22218303

**Published:** 2022-10-29

**Authors:** Chia-Chun Chuang, Chien-Ching Lee, Edmund-Cheung So, Chia-Hong Yeng, Yeou-Jiunn Chen

**Affiliations:** 1Department of Anesthesia, An Nan Hospital, China Medical University, Tainan 70965, Taiwan; 2Department of Medical Sciences Industry, Chang Jung Christian University, Tainan 71101, Taiwan; 3Department of Electrical Engineering, Southern Taiwan University of Science and Technology, Tainan 71005, Taiwan

**Keywords:** multi-task learning, brain–computer interface, steady-state visual evoked potentials, SSVEP signal enhancement, amyotrophic lateral sclerosis

## Abstract

Amyotrophic lateral sclerosis (ALS) causes people to have difficulty communicating with others or devices. In this paper, multi-task learning with denoising and classification tasks is used to develop a robust steady-state visual evoked potential-based brain–computer interface (SSVEP-based BCI), which can help people communicate with others. To ease the operation of the input interface, a single channel-based SSVEP-based BCI is selected. To increase the practicality of SSVEP-based BCI, multi-task learning is adopted to develop the neural network-based intelligent system, which can suppress the noise components and obtain a high level of accuracy of classification. Thus, denoising and classification tasks are selected in multi-task learning. The experimental results show that the proposed multi-task learning can effectively integrate the advantages of denoising and discriminative characteristics and outperform other approaches. Therefore, multi-task learning with denoising and classification tasks is very suitable for developing an SSVEP-based BCI for practical applications. In the future, an augmentative and alternative communication interface can be implemented and examined for helping people with ALS communicate with others in their daily lives.

## 1. Introduction

People with amyotrophic lateral sclerosis (ALS) normally have difficulty communicating with others or devices [1,2,3,4]. A brain–computer interface (BCI) can effectively help people with ALS communicate with others or give commands to external devices without using muscles [4,5,6,7,8,9,10,11]. Compared with other BCIs, steady-state visual evoked potentials-based (SSVEP-based) BCI is one of the most successful interfaces for people with ALS since it has a high signal-to-noise ratio (SNR) [4]. However, in practical applications, the noise of the usage environment greatly reduces the performance of SSVEP-based BCIs. Moreover, for people with ALS, an easy-to-use interface, which can select a predefined vocabulary or phrase by using visual stimuli, would increase users’ acceptance. Therefore, developing a robust SSVEP-based BCI can help people with ALS to effectively communicate with others or devices.

ALS hinders the normal functioning of the motor neurons that control voluntary muscles, causing progressive weakness, muscle twitching, and stiff muscles [9,10,11,12,13]. The abilities to swallow, walk, move hands, and speak are severely degraded. Moreover, the nervous system of affected people is still intact and therefore people with ALS feel pain. Thus, developing a suitable assistive communication interface can effectively help people with ALS.

For BCI applications, the SSVEP-based technologies utilize visual stimuli to induce human brain activity in the central nervous system and then trigger brain potentials [6,7,14]. Via the brain potentials, the SSVEP-based BCIs can easily detect the selected visual stimulus, which represents a predefined vocabulary or phrase. Then, a user can communicate with others or devices without using muscles. Therefore, SSVEP-based BCI is one of the most successful assistive communication interfaces for subjects with ALS [14,15,16,17]. However, the quality of the SSVEP signal is degraded since the SSVEP signals always mix with normal brain activity and background noise. Several researchers have proposed an EEG device with multi-channels and multi-channel-based signal processing to improve the quality of acquired EEG signals [15,16,17]. However, the cost of the SSVEP-based BCIs is increasing and it is uncomfortable for a user having to place a number of electrodes. Thus, developing a robust SSVEP-based BCI using a single channel to operate the BCI is very useful for users.

SSVEP-based BCIs with a single channel have been developed and used to help people with ALS [6,12,15]. However, the performance of these systems is always degraded in practical applications due to the noisy SSVEP signal. Maye et al. proposed an averaging approach to reduce the effects of noise [6]. However, the noise in practical applications is not stable, and therefore the averaging approach cannot effectively remove the noise components. Chen et al. proposed a denoising autoencoder-based SSVEP signal enhancement to deal with the noisy SSVEP signal [12]. For this approach, a denoising autoencoder (DAE) is adopted to learn the characteristics of noise, and then a robust feature is extracted to represent a noisy SSVEP signal. However, the robust feature is used to represent a clean SSVEP signal, not a discriminative feature for a classification task. Therefore, it still has to design a complex neural network structure to achieve an acceptable performance of classification approaches. Generating a robust feature, which can effectively reduce the noisy components of a SSVEP signal and exactly represent the characteristic of a SSVEP signal for the classification tasks, can increase the values of SSVEP-based BCIs for practical applications.

Recently, multi-task learning (MTL) has been proposed to exploit commonalities across tasks to improve generalization by using domain information contained in the training signals [16,18,19,20]. Moreover, MTL uses a shared representation, which is learned for each task, to integrate the different information for each task. For clinical practical applications, MTL can obtain representations that can achieve an acceptable performance of classification and reduce the effects of noise in the SSVEP signal. Thus, MTL is very suitable for developing robust SSVEP-based BCIs.

In this paper, we present a study to develop an SSVEP-based BCI with MTL to help people communicate with others. To assist people in operating the input interface, the visual stimuli with specific frequencies displayed on an LCD monitor are selected as the input interface. To reduce the cost and increase comfort, a single channel is adopted as the input interface of the SSVEP-based BCI. To increase the accuracy of the SSVEP-based BCI in practical applications, MTL is developed to extract the features with denoising and discriminative characteristics.

The rest of this paper is organized as follows. The proposed robust SSVEP-based BCI with MTL is described in Section 2. Section 3 outlines a series of experiments to evaluate the performance of our approach. Conclusions and recommendations for future research are outlined in Section 4.

## 2. An SSVEP-Based BCI with Multi-Task Learning

The flowchart of the proposed SSVEP-based BCI with MTL is shown in Figure 1. Firstly, an LCD monitor is used to display the visual stimuli with different flicking frequencies, and then the elicited EEG signal can be acquired. Secondly, a deep neural network with a multi-task framework is applied to encode the SSVEP signal and to find the designed commands or messages. These procedures are detailed in the following subsections.

### 2.1. SSVEP Signal Acquisition

In this study, a 20” LCD monitor is used to display the visual stimuli, which are five blinking boxes. The five blinking boxes are adopted to represent five commands/messages and are arranged in a regular pentagon for reducing the interference between each visual stimulus. All stimuli are set to a size of 5 × 5 cm with equalized mean pixel luminance and contrast during the presentation. Since the refresh rate of the LCD monitor is 60 Hz, to ensure that the refresh rate is an integer multiple of the presentation frequencies, the blinking frequencies for the five blinking boxes are 6.00 Hz, 6.67 Hz, 7.50 Hz, 8.57 Hz, and 10.00 Hz [7]. Some selected blinking frequencies are not perfect multiples, but it can work as the residual is very small.

The subjects are asked to sit at a viewing distance of 55 cm from an LCD monitor. The elicited SSVEP signal is acquired using a Neuroscan Quickcap electrode cap with 40 channels and a Neuroscan NuAmps™ EEG amplifier. Since the visual stimulation elicits an SSVEP signal in the visual cortex of the brain, only the Oz channel is used to acquire the EEG signal. The reference and ground electrodes are placed at A1 and A2. The sampling rate of the EEG signal acquisition system is 1 kHz. To reduce the computational complexity, the sampling rate of the SSVEP signal is converted to 100 Hz using an anti-aliasing lowpass filter, which is a 19-order Butterworth lowpass filter with a 50 Hz cut-off frequency.

In this study, a visual stimulation procedure with 5 sequences of target stimuli is designed to elicit the SSVEP signal. Each sequence has 3 target stimuli with different stimulation frequencies, which are randomly selected from the given 5 frequencies. Moreover, each visual stimulus with different blinking frequencies is strictly limited to 3 occurrences in a testing procedure. For a testing procedure, the participant is asked to sequentially look at a blinking box by using a speech cue, which refers to the target stimuli in the sequence. The testing procedure is designed as follows.

Step 1: A 5 s countdown delay is designed at the beginning of the testing procedure. For the countdown period, a speech cue is used to ask the participant to look at the corresponding blinking box. Since five blinking boxes are displayed on the LCD monitor, the position of the corresponding target stimulus is also provided to the participant.

Step 2: The participant is asked to look at the target blinking box, and then a series of 10 s of the elicited SSVEP signal is recorded.

Step 3: The participant can rest for 10 s.

Step 4: Step 1 is repeated three times.

Step 5. One minute of compulsory rest is provided for the participant if he/she wants to continue an experiment by using another sequence.

Once the participant finishes a visual stimulation procedure, 3 trials for each stimulus can be obtained. To increase the data size, a trial is then blocked into 10 non-overlapping blocks, with a duration of one second. Hence, for the participant, there are 30 blocks consisting of 100 EEG samples for each target stimulation frequency.

### 2.2. Neural Network with Multi-Task Learning

The framework of the proposed neural network with MTL is shown in Figure 2. In this study, the DAE and SSVEP classification tasks are selected to develop the neural network with MTL. The DAE task is used for SSVEP signal enhancement and reduces the effects of noise in practical applications. The SSVEP classification task is then used to recognize the commands/messages according to the elicited SSVEP signal. By integrating these tasks, the feature encoder can extract the robust feature, which has denoising and discriminative characteristics. The process is introduced in greater detail in the following section.

#### 2.2.1. Embedding Feature Encoder

The neural network structure of the embedding feature encoder is shown in Figure 3. For the embedding feature encoder, an SSVEP signal in the time domain is fed into the input layer. Then, a sequence of convolutional layers, activation functions, and max pooling layers is applied to extract a representation of the input SSVEP signal.

Generally, the following convolutional layer is adopted to perform convolution to the outputs of previous layers, and the weights of previous inputs can be learned. The rectified linear unit (ReLU) is selected as the activation function and used to prevent the exponential growth in the computation required to operate the neural network. The max pooling layer is a pooling operation that calculates the maximum value for patches of a feature map. Then, the computational complexity can be effectively reduced by using downsampling features.

Finally, a fully connected layer is adopted to extract the embedding features, which are the bottleneck features of the network. The fully connected layer applies a linear transformation to the input vector through a weights matrix, and then a bottleneck feature can be obtained. The bottleneck feature is also called the embedding feature and is used to represent the input SSVEP signal. Moreover, using MTL, the embedding features will contain denoising and discriminative characteristics.

#### 2.2.2. Denoising Task

The neural network structure of the denoising task is shown in Figure 4. Firstly, a fully connected layer is applied to decode the encoding features. Secondly, decoded features are reshaped to a set of features and the size is the same as the kernel of the convolution layer in the encoder. Finally, a sequence of deconvolutional layers and up-pooling layers is applied to reconstruct the SSVEP signal.

In this study, a DAE can be developed by combining the structures of the embedding feature encoding and the denoising decoder. For practical SSVEP-based BCIs, the elicited SSVEP signal always contains different degrees of noise, which are defined as non-target frequency components. Thus, an acquired SSVEP signal, *x*(*t*), is defined as
*x*(*t*) = *s*(*t*) + *n*(*t*),(1)
where *s*(*t*) and *n*(*t*) are an ideal SSVEP signal and a noise signal, respectively. Since the ideal SSVEP signal cannot be obtained, the ideal SSVEP signal of different visual stimuli is assumed as a sine wave and defined as
(2)$$s_{i}\left(t\right)=A\sin\left(2\pi f_{i}t+\varphi_{i}\right)$$
where *i* is the *i*-th visual stimulus. *A*, *f_i_*, and *φ_i_* are the amplitude, ordinary frequency, and phase, respectively.

In the training stage, *f_i_* can be obtained via the frequency of the corresponding visual stimulus. Generally, there is a phase shift, *φ_i_*, between the acquired SSVEP signal and an ideal SSVEP signal. Therefore, the cross-correlation is applied to find the *φ_i_* between the input SSVEP signal and a sine wave, the phase of which is zero. According to the estimated phase shift, an ideal SSVEP signal, the phase of which is equal to the input SSVEP signal, can be generated. The mean square error is used as the loss function of the denoising decoder, *L_D_*.

#### 2.2.3. Classification Task

The neural network structure of the classification task is shown in Figure 5. Firstly, a fully connected layer is applied to decode the embedding feature. Secondly, the decoded feature is reshaped to a set of features and the size is the same as the kernel size of the convolution layer in the encoder. Thirdly, a sequence of convolution layers, activation functions, and max pooling layers is adopted to extract reasonable features for classification. Finally, a sequence of fully connected layers is used to find the recognition results.

In the training stage, the cross-entropy function is selected as the loss function of the classification task and denoted as LC. To estimate the parameters of the proposed neural network, as shown in Figure 1, a loss function by weighting LD and LC is used and defined as
*L*(*x*) = *LC*(*x*) + *αLD*(*x*),(3)
where *α* is a weighting factor. The *α* is used to balance the effects of loss for DAE and classification tasks. The effects of the DAE and classification tasks are assumed to be equal, and then it is set by referring to the distributions of the loss score for DAE and classification tasks.

### 2.3. The Selected Hyperparameters of the Proposed Neural Networks

Moreover, the hyperparameters of the proposed neural network with MTL are shown in Table 1. The Adam algorithm, which is an adaptive learning rate optimization technique, is selected as the optimizer for training neural networks. The learning rate, number of iterations, and batch size are 0.001, 200, and 32, respectively. Furthermore, *α*, which refers to the distributions of the loss score for denoising and classifying decoders, is 0.08 in this study.

## 3. Experimental Results and Discussions

The performance of the proposed SSVEP-based BCI with MTL is evaluated in the following section.

### 3.1. Experimental Environment

In this study, 24 health participants (18 males and 6 females) aged between 21 and 23 were asked to participate in the experiment. The participants gave informed consent to the study conditions and were able to abort the experiment if they felt uncomfortable. The participants did not have epilepsy or previous experience using the SSVEP-based BCIs. For a user, it is very useful and convenient to use a system without any adaptation/training process. To find an embedding feature, which is user-invariant, a user-independent SSVEP-based BCI can be developed. Therefore, leaving-one-participant-out cross-validation, where the number of folds equals the number of participants in the data set, was adopted to evaluate the proposed approaches. Thus, the learning algorithm of the proposed approach was applied once for each participant, using all other participants as a training set and using the selected participant as a single-item test set.

### 3.2. Experimental Results of Denoising Task

In this sub-section, the SNR is applied to evaluate the performance of the denoising task. For each input SSVEP signal, the sine wave, the phase of which is equal to the input SSVEP signal, is selected as the ideal SSVEP signal. To compare the performance of the MTL approach, the DAE, which has the same hyperparameters of the neural networks used for embedding the features encoder and denoising decoder, is adopted as the baseline system, and the experimental results are shown in Table 2. The average SNR for original SSVEP signal, DAE without MTL, and DAE with MTL are 2.50 dB, 6.23 dB, and 6.86 dB, respectively. It is clear that the noise components increase with frequency in this dataset. The DAE can effectively reduce the noise components. Moreover, the SNR of DAE can be improved again by integrating the classification task. The embedding feature of DAE with MTL will preserve the properties of classification results. Therefore, the proposed DAE with MTL can suppress the noise components for practical applications.

To examine the effects of noise components for practical applications, the canonical correlation analysis (CCA), which has been widely used in SSVEP-based BCIs [21], is adopted in this experiment. The number of harmonics for CCA was set to be 4, and the experimental results are shown in Table 3. The average accuracy and standard deviation for original SSVEP signal, DAE without MTL, and DAE with MTL are 87.45% ± 1.86%, 90.14% ± 1.19%, and 90.70% ± 1.24%, respectively. The Wilcoxon signed-rank test is then selected to test whether or not there is a significant difference between two population means. The *p*-values for comparing DAE with MTL vs. DAE and DAE with MTL vs. original SSVEP signal are 0.03662 and 0.00138, respectively. The statistical results are significant at *p* < 0.05, meaning that the proposed approach outperforms other approaches. Referring to Table 2, it is clear that the recognition rate increases with SNR. Hence, suppressing noise components can effectively increase the recognition rate of an SSVEP-based BCI.

### 3.3. Experimental Results of Classification Task

In this sub-section, the performance of the classification task by using the proposed neural networks with MTL is evaluated. A neural network, which has the same hyperparameters of the neural network used for embedding the feature encoder and classification decoder, is selected as the baseline system. The experimental results are shown in Table 4. The average accuracies and standard deviation for baseline system and neural network with MTL are 89.70% ± 1.63% and 93.44% ± 1.14%, respectively. The Wilcoxon signed-rank test is also selected to test the results of the classification approach by using neural networks with and without MTL. The *p*-value is 0.02088 and is significant at *p* < 0.05. Therefore, the neural network with MTL can effectively improve the performance of neural network-based classification. Referring to Table 3, the accuracy of the baseline system is greatly affected by the noise. However, the accuracy of the neural network with MTL can effectively reduce the effects of noise. This demonstrates that the embedding feature with denoising characteristics is highly effective for suppressing noise.

To detail the characteristics of the proposed SSVEP-based BCI with MTL, the architecture of the DAE approach [12] is selected. In this approach, the encoder and decoder networks are trained, and then the weights of the encoder network are fixed. Then, the neural networks for the classification task, including four convolution layers and two fully connected layers, are used to connect the encoder network. In the training stage, we only update the weights of the neural network for the classification decoder. This approach is denoted as DAE_baseline.

To objectively compare the denoising approach, a two-stage training approach is also used. In the first training stage, the whole weights of the neural network with MTL are updated. In the second training stage, the weights of the embedding feature encoder are fixed and we only update the weights of the neural network of the classification decoder. This approach is denoted as the updated neural network with MTL.

The accuracy and standard deviation for the denoising approach, DAE_baseline, the neural network with MTL, and the updated neural network with MTL approach are 92.92% ± 1.05%, 93.44% ± 1.19%, and 94.64% ± 1.59%, respectively. The result of the neural network with MTL is similar to that of the DAE_baseline, and the effects of noise components can be effectively reduced. Moreover, to achieve an acceptable performance for practical applications, a two-stage training approach is very useful to increase the accuracy of MTL for SSVEP-based BCIs.

Comparing the results in Table 3 and Table 4, the accuracy of CCA with SSVEP signal enhancement by using DAE with MTL and neural network-based classification by using DAE with MTL are 90.70% and 93.44%, respectively. The results show that the proposed approach outperforms CCA, and can achieve an acceptable performance for practical applications. Moreover, the embedding feature of DAE is forced to find a common stimulus representation that works for all participants, and the classification network is optimized for the current participant only. Therefore, MTL with DAE and SSVEP classification tasks can successfully improve the performance of the SSVEP-based BCIs. With the acceptable performance, an augmentative and alternative communication interface can be designed to express some intents that are selected from a predefined set, including specified vocabularies or phrases. Therefore, people with ALS can use the predefined vocabularies or phrases to communicate with others in their daily lives.

## 4. Conclusions

In this study, a robust SSVEP-based BCI is successfully developed to help people with ALS communicate with others by using visual stimulation. MTL can integrate the advantages of denoising and classification tasks. The denoising task can effectively suppress the noise components in SSVEP signals and then robust embedding features can be extracted to improve the performance of classification. The classification task is adopted not only to find a suitable recognition result of SSVEP-based BCI but also to provide classification information for improving the performance of the denoising task. The experimental results show that the proposed neural networks with MTL outperform other approaches. Thus, the denoising and the classification tasks have complementary effects on the recognition rate of an SSVEP-based BCI. Therefore, MTL with denoising and classification tasks is suitable for developing an SSVEP-based BCI for practical applications. In this study, the convolutional layers are adopted as the basic units of MTL. Recently, many novel neural network structures, such as Inception, DenseNet, and SENet, are proposed, and these novel neural networks can be easily adopted as the basic units of our approach. The proposed approaches can also be used to implement an augmentative and alternative communication interface so that people with ALS can evaluate the suitability of the proposed SSVEP-based BCI.

## Figures and Tables

**Figure 1 sensors-22-08303-f001:**
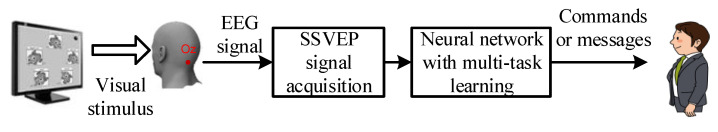
The flowchart of the proposed SSVEP-based BCI with MTL includes a visual stimulation procedure, SSVEP signal acquisition, and neural network with MTL.

**Figure 2 sensors-22-08303-f002:**
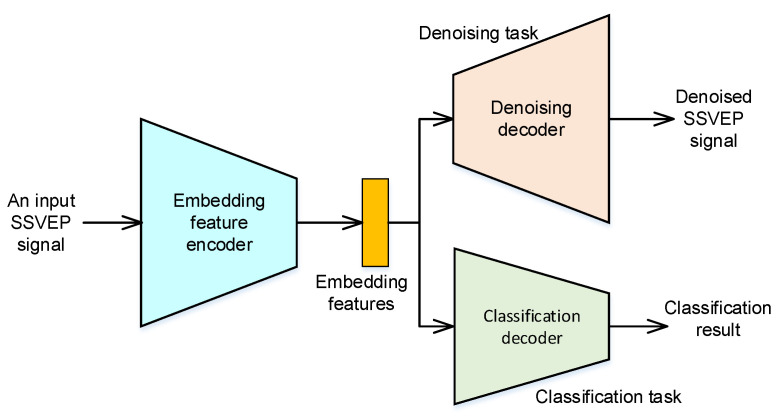
The framework of the proposed multi-task based neural network.

**Figure 3 sensors-22-08303-f003:**
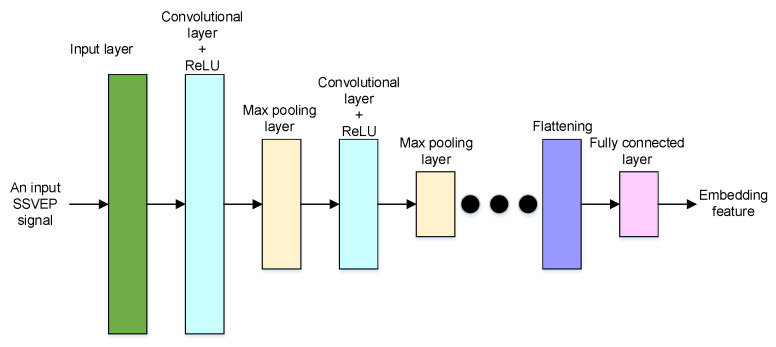
The neural network structure of the embedding feature encoder.

**Figure 4 sensors-22-08303-f004:**
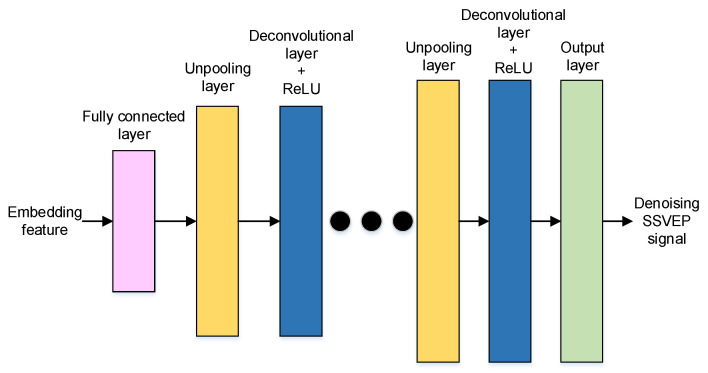
The neural network structure of the denoising decoder.

**Figure 5 sensors-22-08303-f005:**
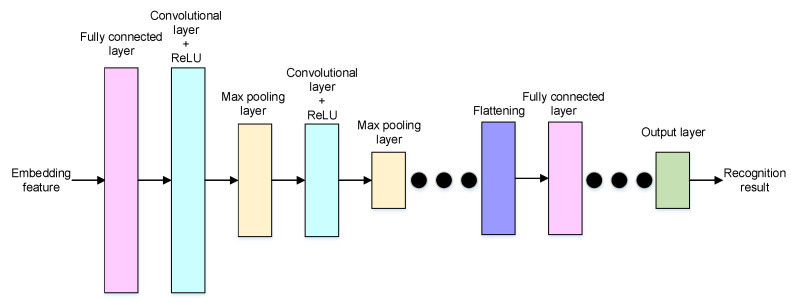
The neural network structure of the classification decoder.

**Table 1 sensors-22-08303-t001:** The hyperparameters of the proposed neural network with MTL (I: input layer, C: convolutional layer, M: max pooling layer, F: fully connected layer, U: up pooling layer, DC: deconvolutional layer, S: sigmoid layer, NS: the number of nodes, FS: filter size, KS: kernel size).

Feature Encoder			Denoising Decoder			Classification Decoder		
Layer	Hyper- Parameter	Value	Layer	Hyper- Parameter	Value	Layer	Hyper- Parameter	Value
I	NS	100	F	NS	200	F	NS	100
C	FS, KS	8, 16	U	KS	2	C	FS, KS	16, 16
M	KS	2	DC	FS, KS	8, 8	M	KS	2
C	FS, KS	8, 8	U	KS	2	C	FS, KS	16, 8
M	KS	2	DC	FS, KS	8, 16	M	KS	2
F	NS	128	S	NS	100	F	NS	64
						F	NS	32
						S	NS	5

**Table 2 sensors-22-08303-t002:** The average SNR (dB) for original SSVEP signal, DAE without MTL, and DAE with MTL.

	Frequency of Stimuli				
	6.00 Hz	6.67 Hz	7.50 Hz	8.57 Hz	10.00 Hz
Original SSVEP signal	3.99	1.85	3.61	1.64	1.43
DAE without MTL	7.18	5.14	6.98	5.42	4.42
DAE with MTL	7.13	6.26	7.22	6.36	5.35

**Table 3 sensors-22-08303-t003:** The accuracy (μ ± σ%) of CCA for original SSVEP signal, DAE without MTL, and DAE with MTL.

Frequency of Stimuli	Original SSVEP Signal	DAE without MTL	DAE with MTL
6.00 Hz	91.11 ± 2.07	91.78 ± 2.19	91.50 ± 2.25
6.67 Hz	86.08 ± 3.20	89.97 ± 3.86	89.83 ± 2.45
7.50 Hz	89.56 ± 3.57	91.64 ± 2.89	92.33 ± 3.10
8.57 Hz	87.06 ± 3.35	89.00 ± 4.39	90.11 ± 3.76
10.00 Hz	84.42 ± 3.48	88.31 ± 2.59	89.69 ± 3.59

**Table 4 sensors-22-08303-t004:** The accuracy (μ ± σ%) of SSVEP-based BCI using neural network-based classification with and without MTL.

Frequency of Stimuli	Neural Network without MTL	Neural Network with MTL
6.00 Hz	92.64 ± 2.37	94.58 ± 1.91
6.67 Hz	89.03 ± 3.10	92.50 ± 2.43
7.50 Hz	91.25 ± 2.50	95.00 ± 1.50
8.57 Hz	88.06 ± 3.92	92.78 ± 3.33
10.00 Hz	87.50 ± 3.83	92.36 ± 3.09

## Data Availability

Not applicable.

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
