# Peer review of "Multi-Task Learning-Based Deep Neural Network for Steady-State Visual Evoked Potential-Based Brain–Computer Interfaces"

_sensors, 2022, doi:10.3390/s22218303_

Round 1
Reviewer 1 Report
In this paper, the authors propose a single channel-based SSVEP-based BCI system with the multi-task learning. This system can suppress the noise components and obtain high classification accuracy. Experimental results are provided to validate the advantage of the proposed algorithm. Overall, the study is both complete and convincing. I have the following concerns:
1. Line 11-15: The description of the fundamental problems can be condensed. The main work should be point out.
2. Introduction: Lots of newly EEG-based BCI signal processing method [1] using the Riemannian manifold [2] can be added to enrich this part.
[1] P. L. C. Rodrigues, C. Jutten and M. Congedo, "Riemannian Procrustes Analysis: Transfer Learning for Brain–Computer Interfaces," in IEEE Transactions on Biomedical Engineering, vol. 66, no. 8, pp. 2390-2401, Aug. 2019, doi: 10.1109/TBME.2018.2889705.
[2] X. Hua, Y. Ono, L. Peng and Y. Xu, "Unsupervised Learning Discriminative MIG Detectors in Nonhomogeneous Clutter," in IEEE Transactions on Communications, vol. 70, no. 6, pp. 4107-4120, June 2022, doi: 10.1109/TCOMM.2022.3170988.
3. Line 54 should be deleted.
4. More comparison results on classification networks, e.g., Inception, DenseNet, SENet, should be provided to validate the superiority of the proposed method.
5. Conclusion: The authors missed the further improvements and the potential drawbacks of the proposed method.
Author Response
Dear Reviewer of MDPI Sensors,
In this document, we describe how we address the reviewers’ comments in the revised manuscript (original Manuscript ID: sensors-1909354). We made a response to each comment item by item in detail. We hope that this revision can address all the editorial concerns. We would like to thank the reviewers for their concrete and constructive comments in helping us to improve the presentation of the manuscript.
Sincerely,
Yeou-Jiunn Chen

Reviewer 2 Report
The paper introduced an interesting approach employing multi-task learning to detect and denoise an EEG signal. In the EEG signal, the frequency of SSVEPs can be detected more accurately compared to an approach that omits multi-task learning.
The work is very interesting but in my opinion, could benefit from an iteration with the goal to improve the overall presentation.
For instance, in Section 3.1 it is difficult to understand how the stimuli were presented to the study subjects. In Fig 1 the authors present a pentagon of stimuli and in Section 3.1 the authors state that there are 3 stimuli. Were these stimuli presented at the same time? Or did the subjects focus on them one after the other? I believe the latter is true but it could be described in more detail. Further, the position of the stimuli could be explained. Were they shown at the same position as shown in Figure 1 (pentagon) or in the middle of the screen?
I believe the work needs an extra iteration before publishing. In the following, I present improvement suggestions.
==Introduction==
The introduction makes some claims that could be supported by the literature. For example in Line 30 "Comparing..." Further, the authors state that a "simple interface would increase user acceptance. The authors could consider stating an example of why this could be the case. What is needed by ALS patients in this context? That would help to motivate the work.
One thing that stood out to me was reference 11 (Line 39). Although it is difficult to access the corresponding paper, I doubt that this reference is relevant to the sentence beginning in Line 37. The authors should consider checking this reference.
In Line 40 the authors state that people feel pain as they have "normal thinking abilities". Maybe this could be rephrased a bit. Isn't it more like the nerve system that is still intact and therefore patients feel pain? I don't think that the ability to think induces the pain here.
In Line 44, the authors introduce the following claim: "According to the brain potentials, the SSVEP-based BCIs can easily detect the intentions of a user." I have some difficulties with this statement. How do SSVEPs convey the intention of users? They indicate that the user focuses on a stimulus flickering at a specific frequency. Or can the authors provide a reference for this statement? But to my knowledge, intentions are not encoded in SSVEPs. One can simply stare at a stimulus with no intention and we could still detect the corresponding frequency. I suggest deleting this statement.
Line 47: is there a reference for this statement? (see additional literature below). Also, in Line 55 a reference should be considered.
In Line 49 the authors talk about normal brain activity and complex background noise. What is complex background noise? Isn't it just background noise that is there all the time? My suggestion would be to omit the word "complex".
Line 54: "SNR (SSVEP signal Enhancement)" is this a heading of a paragraph? It seems to me like the authors have forgotten to delete this line.
Line 58: "not good enough" should be rephrased. E.g., "the approach has its limitations" Also, the authors could state the corresponding reasons. Is there still too much noise?
Line 74: "In this study,..." I would rephrase that. The study comes out of the blue. I suggest writing. In this paper, we present a study/evaluation. Also in Line 191, the authors talk about a "test of the project". I suggest sticking to one term that describes the conducted work best. In this case, I believe "study" is the most accurate. Additionally, the authors could write: "the participants gave informed consent to the study conditions" Further, the authors state that the "data was collected in three days" I believe this could be omitted as it does not seem relevant for the introduced research.
==2 Multi-task learning based SSVEP-based BCI==
In my opinion, this heading used the word "based" too often. The authors could consider rephrasing it a bit. Also, it could follow the APPA style of headings. Further, the authors guide the reader through the paper using references to "subsections". I suggest just writing something like "the approach is introduced in greater detail in the following."
Line 102: Technically not all frequencies are perfect multiples but in my opinion, it should work as the residual is very small.
Line 125: How is the EEG input shaped? How long are the samples or sample array/vector passed to the network? Are they in the time domain or where they transformed prior to getting fed into the NN?
Line 172: The authors could elaborate a bit more on the weight (alpha). How was it picked? How big was alpha?
Line 178: The authors state that the frequencies were selected randomly. How does this ensure that each frequency is used as often as the others? This might impact the general outcome of the results. Did the authors ensure that there was an even distribution among the fs?
==3 Experimental Results and Discussion==
In Line 187 the authors mention one-second blocks consisting of 100 samples. I believe that this is the input then (Line 125). I suggest that this information could be consolidated into a paragraph in the previous section that describes the approach.
In the discussion, the authors, unfortunately, do not discuss the implications for ALS patients. How do the results help these patients in their everyday lives? Is denoising in the context of ALS more important than other domains in which SSVEP can be employed? Currently, the motivation of helping ALS patients is strongly emphasized in the beginning but in the end not mentioned anymore.
==Conclusison==
Similar here, in the introduction, the paper focuses a lot on ALS patients but in the end, there are no implications given that this research can have on their lives. What does the approach provide in the context of ALS? The authors state that BCIs that are simple to deploy are needed. In this sense, how does the approach help to provide BCI systems suitable for ALS patients? It is correct that a better classification can be used by ALS patients to communicate but in the context of ALS the authors could emphasize why we need such a BCI system. For example, to allow for easy deployment.
Other than that the overall approach has merit in my opinion. It seems like a reasonable incremental step of existing approaches ([10]). Nonetheless, I believe that the presentation clarity could improve the paper and help to convey its message in a second iteration.
##Additional Comments##
==Structure==
The structure of the paper could be improved. For example, details on the study are a bit scattered across sections and could be introduced altogether in the dedicated evaluation or study section. Also, there is no dedicated related work section. I believe the authors could have introduced a wider array of literature that supports this work.
The approach is a bit hard to comprehend until the authors introduce that they used the approach presented in reference 10. This could be stated earlier when the approach is introduced in Section 2.2 for example. Or the authors could emphasize this a bit more in Line 60.
==Ethicals==
Did the authors check if the subjects of the study had difficulties with the perception of flickering stimuli? E.g., epilepsy. The authors state that they were healthy but did they check beforehand if the participants suffer from seizures? Did the participants know if they could abort the experiment?
##Additional Literature##
https://ieeexplore.ieee.org/abstract/document/9283310
https://ieeexplore.ieee.org/abstract/document/7335644
Minor comments:
- The paper requires proof-reading
- noises -> noise
- DAE: this abbreviation is never introduced.
- MTL: the authors could use this abbreviation throughout the paper
- Typo: Line 64 - effetely
- Typo: Line 84 section 4 -> Section 4 (If there is a number after section, section should be uppercase)
- Line 43: SSVEP techonlogies -> SSVEP-based techonlogies
- Figure 1: Caption: the caption provides very little information in addition to the image. I suggest guiding the reader through the image by explaining the various parts of the figure. The authors could consider that for the remaining figures as well.
- Typo: Line 126: SVEP -> SSVEP
- Typo: Line 190: experiments -> experiment (or was there more than one?)
- Line 200: In this subsection,... -> In this section,...
- Typo: Line 202: SSEVP -> SSVEP
Author Response

(The authors gave the same response as above.)

Reviewer 3 Report
The authors propose a new artificial neural network architecture for the detection of steady-state visual evoked potentials (SSVEPs). Their architecture is novel by making use of an embedding that is shared for two tasks: denoising EEG and classification. The authors show that this multi-task learning leads to a higher decoding accuracy. Although these results are valuable to the brain-computer interfacing field, I feel that the manuscript is currently not yet in a publishable state. Specifically, (1) several important details are missing in the description of the methods harming the overall reproducibility of this work, (2) the claims of improved decoding performances are done with absolute averages but should instead be made on the basis of proper statistical tests, and (3) the manuscript contains many typos and language errors that can easily be improved. Please see my detailed comments below.
Major
· Throughout the manuscript, predominantly the methods section, many important details seem missing. Firstly, there is limited information presented about the parameters of the neural network, which prevents replication. How many layers were used (figures 3-5 contain dots to visualize there are multiple, but nowhere it is stated how many)? Within convolutional layers, how many kernels were used and what were their kernel sizes? Within pooling layers, what were the kernel sizes? Within dense layers, what were the number of units? When training the network, how much data was used for training, how much for validation, and how much for testing? When training the network, which optimizer was used (Adam?), what was the learning rate, did you use a learning rate scheduler, did you use any type of regularization, when did you stop learning (e.g., number of iterations or early-stopping)? In equation 3, the weighting factor alpha is introduced that weights the two multi-task losses: how is this parameter set/optimized? Secondly, I did not fully understand the experimental protocol. Did I understand correctly that per participant only 10 trials were collected, each of 10 seconds, with 1 minute plus 10 seconds inter-trial interval? Why only so few trials, and why did you incorporate such a long inter-trial interval? How were trials cued, were they in random order? Since there are 5 targets, does this mean there are 2 trials per class?
· Related to the previous point, a standard article structure could improve finding relevant information in the manuscript. Typically, the methods section contains an initial section on the participants which is now hidden in the results section page 6 line 189. Subsequently, the data is described with the raw configuration as well as the preprocessing steps, which are again scattered throughout the current manuscript. Subsequently, the experimental pipeline is described. Finally, the analysis methods are described. Such a standard structure could substantially improve the state of the manuscript.
· As mentioned in the manuscript, 24 participants were recorded. Without any further mentioning, it is stated that a leave-one-participant-out cross-validation is performed (page 6 line 193). This means this study is involved in transfer-learning, which is another ballpark than within-subject classification, which tends to be “easier”. Why did you choose for such transfer-learning classification? Please add this motivation also to your introduction and discussion, and discuss relevant literature thereof to put your study in the related field.
· Related to the previous point, tables 2-4 report absolute grand average decoding performances. Given these values, statements are made that the one method performs better than the other. In my honest opinion, this needs to be backed-up by proper statistics (e.g., Wilcoxon paired test). This is motivated by the effect sizes in table 4 that are relatively small. If in these cases the variance over the 24 participants is relatively large, these reported small differences might actually be “simple” random fluctuations and not an actual “true” effect. Please add proper statistics to back-up your statements. I would also suggest adding a results figure, for instance a bar/box plot, that shows both the means/medians as well as the variance/range.
· Many typos and grammar mistakes are found throughout the manuscript. Please carefully proofread the manuscript, see the minor points below for a non-exhaustive list.
· Finally, although not necessary for the scope of this article and journal, I would like to see how your method compares to the state-of-the-art in SSVEP decoding. At least this could be done qualitatively in the discussion. For instance, see the recent work by Zhang and colleagues, who compared their proposed method (also a neural network) to several commonly used methods such as CCA, TRCA, Conv-CA, etc.: Zhang, X., Qiu, S., Zhang, Y., Wang, K., Wang, Y., & He, H. (2022). Bidirectional Siamese correlation analysis method for enhancing the detection of SSVEPs. Journal of Neural Engineering, 19(4), 046027.
Minor
· Title: replace “to” with “for”
· Abstract/introduction: you explicitly state “Thus, a robust SSVEP-based BCI is developed to help people with ALS”. Instead, you tested with healthy participants. Please rephrase these statements throughout the manuscript to prevent the misunderstanding that you did test with ALS patients.
· Abstract line 20: replace “outperformance” with “outperform”
· Introduction line 27: replace “is” with “are”
· Introduction line 28: replace “The” with “A”
· Introduction line 37: replace “ASL” with “ALS”
· Introduction line 44L replace “wavers” with “waves”
· Introduction line 44: you state “stimuli to induce human brain waves and then trigger brain potentials”. What is the difference between brain waves and brain potentials?
· Introduction line 44: suggestion to replace “According” with “Via”
· Introduction line 50: here you state that you use multi-channel EEG. However, in the methods you mention you only use Oz and ignore all other channels. Please prevent these misconceptions.
· Introduction line 54: redundant line?
· Introduction: there are many typos in the use of the word “noise”, spelled as “noising” or “noises”. The word “noise” is already plural.
· Introduction line 74: again a statement on ALS, please refrain or rephrase carefully.
· Introduction line 76: suggestion to replace “to easy for” with “to assist”
· Section 2.2: question: would it make sense to use the denoised EEG as input to the decoder?
· Section 2.2.1 line 126: “SVEP” should be “SSVEP”
· Section 2.2.1 line 127: you state a convolution over the channel dimension, but this dimension only contains one channel (Oz)? What is this layer doing?
· Line 147: “combing” should be “combining”
· Line 157: you state “cross-correlation” for fitting the phase of the target sine-wave. Do you mean cross-validation? If so, please explain on which part of the (training) data this was done. If not, please explain what this means.
· Figure 5: the text mentions that there is a dense layer before the convolutional layers (line 160-161). I do not see this in the figure?
· Equation 3: how is alpha set/optimized?
· Section 3.1: You mention that each of the 5 stimuli are sequences of 3 of the different frequencies. Why did you choose this rather unusual stimulus design (as opposed to using one frequency per stimulus)? Additionally, does this mean that throughout a 10-second trial, the first 3.3 seconds were frequency 1, the second frequency 2, etc.? How are these stimuli designed exactly (and why)?
· Section 3.1: why do you add 10 seconds rest and additionally 1 minute rest?
· Line 192: You say the data were collected in three days. Do you mean that all participants were recorded on three days, or that the data of one participant were recorded over three days?
· Line 202: “SSEVP” should be “SSVEP”
· Section 3.2: You propose a denoising network, which is really nice. You have several repetitions of similar trials. Could you maybe compare your (results of the) denoising algorithm to grand average time-series of the stimuli you present? An average “denoises” a signal with sqrt(n) where n is the number of repetitions. Maybe useful as comparison?
· Line 215: “canonical correspondence analysis” should be “canonical correlation analysis”
· Lines 248-249: abbreviations MTL and DAE are not introduced.
Author Response

(The authors gave the same response as above.)

Round 2
Reviewer 1 Report
The authors have addressed all my concerns.
Author Response
Detailed Responses to the Reviewers’ Comments:
The authors have addressed all my concerns.
Response:
We appreciate the time and effort the reviewer has dedicated to providing valuable feedback on the manuscript. Thank you very much.

Reviewer 3 Report
The authors have substantially improved their manuscript. I still have one major concern regarding the analysis and with that the interpretation of the results, which concerns the cross-validation procedure. Additionally, I have one major concern that might substantially improve readability, and only some minor proposed changes.
Major:
· I still do not understand the leave-one-subject-out (LOSO) cross-validation CV. Typically, LOSO is used for transfer-learning: with N participants, a single model is trained on N-1 participants, and that single model is evaluated on the left-out participant. This procedure is done N times, once with each participant as test data. This, by definition is transfer-learning, because the model needs to generalize to a new unseen participant. Is this what you are doing? Contrary, what you might be doing is keeping a hold-out set, where N-1 participants are used to tweak hyper-parameters, and only at the end you analyze the Nth participant. This however you only do once, as you have only one hold-out set. Is this what you are doing? If so, please, you must mention for each of your results that these are of N=1 participant! Additionally, which participant was this last participant? Is it the last one, was this decided before starting analysis (i.e., not cherry-picking)? If you did anything else, please explain carefully.
Given that you have results over participants (otherwise statistics would not make sense), I assume that you are either in the transfer-learning regime, or doing something absolutely different, which one should not call LOSO CV. In the case you did the hold-out analysis, you should clearly mention this as all your results are based on N=1, as well as your statistical tests are invalid, as they would be done on one value only? Please explain.
· Section 3.1 and table 1 belong in the methods section (section 2). Please add these phrases to the correct sections.
Minor:
· Page 3, line 136-139: consider using blocks that contain runs that contain trials. Segments, to me, sounds like a subset of a trial. Importantly, on line 139, do not use the word samples, as I think you mean trials (of 10 seconds). Samples is ambiguous, as it may refer to the EEG samples within a trial. Please stick to the terminology that is generally accepted within the field.
· Page 6, line 207: please consider adding a line on how alpha is set/optimized. If it is explained elsewhere, refer to that section.
Author Response

(The authors gave the same response as above.)

Round 3
Reviewer 3 Report
The authors have improved again their manuscript. The final issue is still on the description of the carried out analysis. My apologies to the authors if I was not clear in my previous rounds of feedback. What I am trying to do is to make sure that readers understand clearly what has been done in this study (replicability).
I am convinced you performed a leave-one-subject-out (LOSO) cross-validation (CV). And yes, this is a standard CV scheme to use within transfer-learning for BCI, e.g. see this comprehensive review: Jayaram, V., Alamgir, M., Altun, Y., Scholkopf, B., & Grosse-Wentrup, M. (2016). Transfer learning in brain-computer interfaces. IEEE Computational Intelligence Magazine, 11(1), 20-31.
I would therefore suggest you stick with LOSO CV in your description of your methods. If you were to stick with leave-one-out CV, this might indicate leave-one-trial-out CV, which is a within-subject (so no transfer-learning) CV scheme, which is more typically used within the BCI field, but is *not* what you have done. Again, please use LOSO CV in your description.
This does mean that you did transfer-learning. Yes, transfer-learning is a method to generalise from one task to another task. This "task" here is generalising from one pool of participants to a new participants, i.e., to transfer information from one participant to another. Transfer-learning (or cross-participant decoding) is therefore much more difficult than within-subject decoding, simply because the changes in data distributions are expected to be larger for cross-subject (transfer-learning) than for within-subject. This terminology has been accepted in the BCI field, see again the review.
Now my big question is, why did you use this cross-subject analysis (and not a more standard within-subject analysis)? If you were interested in merely comparing methods (and not transfer-learning), it would have been easier to do this in a within-subject analysis. However, I can imagine that the embedding learning becomes very important for cross-participant decoding, simply because now you need to find an embedding that is subject-invariant. To this end, one could maybe hypothesise that this is why in your study the embedding learning seems to work, simply because it is forced to find a common stimulus-representation that works for all participants with the denoising network, while the classification network is of course optimised for only the current participant. So please add a motivation for this analysis in your introduction/methods, and maybe a statement on the question whether this would also work for within-subject analysis in the discussion.
To summarise:
- Please again use leave-one-subject-out (LOSO) cross-validation (CV).
- Please provide a motivation and discussion for why you chose this cross-participant analysis over a more simpler within-subject CV scheme.
- Please add to the tables e.g. the standard deviations apart from the means. Alternatively, prove bar-graphs with error-bars to reflect e.g. standard errors apart from the mean, or box plots with means/medians and inter-quartile ranges.
Author Response

(The authors gave the same response as above.)
